# Purkinje Cell Activity Resonation Generates Rhythmic Behaviors at the Preferred Frequency of 8 Hz

**DOI:** 10.3390/biomedicines10081831

**Published:** 2022-07-29

**Authors:** Staf Bauer, Nathalie van Wingerden, Thomas Jacobs, Annabel van der Horst, Peipei Zhai, Jan-Harm L. F. Betting, Christos Strydis, Joshua J. White, Chris I. De Zeeuw, Vincenzo Romano

**Affiliations:** 1Department of Neuroscience, Erasmus MC, 3015 GD Rotterdam, The Netherlands; stafbauer@gmail.com (S.B.); n.vanwingerden@erasmusmc.nl (N.v.W.); t.jacobs@erasmusmc.nl (T.J.); annabelvanderhorst@gmail.com (A.v.d.H.); p.zhai@erasmusmc.nl (P.Z.); j.betting@erasmusmc.nl (J.-H.L.F.B.); c.strydis@erasmusmc.nl (C.S.); j.white@erasmusmc.nl (J.J.W.); c.dezeeuw@erasmusmc.nl (C.I.D.Z.); 2Department of Quantum & Computing Engineering, Delft University of Technology, 2628 CD Delft, The Netherlands; 3Netherlands Institute for Neuroscience, Royal Academy of Arts and Sciences, 1105 BA Amsterdam, The Netherlands

**Keywords:** Purkinje cell, cerebellum, theta resonance, whisker, optogenetics stimulation

## Abstract

Neural activity exhibits oscillations, bursts, and resonance, enhancing responsiveness at preferential frequencies. For example, theta-frequency bursting and resonance in granule cells facilitate synaptic transmission and plasticity mechanisms at the input stage of the cerebellar cortex. However, whether theta-frequency bursting of Purkinje cells is involved in generating rhythmic behavior has remained neglected. We recorded and optogenetically modulated the simple and complex spike activity of Purkinje cells while monitoring whisker movements with a high-speed camera of awake, head-fixed mice. During spontaneous whisking, both simple spike activity and whisker movement exhibit peaks within the theta band. Eliciting either simple or complex spikes at frequencies ranging from 0.5 to 28 Hz, we found that 8 Hz is the preferred frequency around which the largest movement is induced. Interestingly, oscillatory whisker movements at 8 Hz were also generated when simple spike bursting was induced at 2 and 4 Hz, but never via climbing fiber stimulation. These results indicate that 8 Hz is the resonant frequency at which the cerebellar-whisker circuitry produces rhythmic whisking.

## 1. Introduction

The olivocerebellar system is essential for sensory-motor processing during the fine-tuning of motor behaviors [1,2,3,4,5,6,7,8,9,10,11,12,13,14,15,16,17]. Conversely, a dysfunctional cerebellum results in diseases characterized by loss of coordination, such as spinocerebellar ataxia [4,18,19,20]. To attain smooth, coordinated movements, cerebellar neurons must operate at specific frequency bands that are decoded by extracerebellar brain areas conveying inputs to various motor domains [21]. Some brain areas receiving the cerebellar output, can, in turn, feedback signals into the olivocerebellar system, creating a series of recurrent loops that could sustain rhythmic activity [22,23,24,25].

Several studies have focused on the preferred frequency band at which cerebellar input sources, the mossy and climbing fiber systems, operate. Mossy fibers originate from many areas of the central and peripheral nervous system and terminate in the input layer of the cerebellar cortex, the granular layer [26,27]. In this layer, granule cells fire action potentials in bursts with oscillatory and resonant properties within the theta-band [28,29,30,31]. Such properties are thought to regulate plasticity and gate sensory-motor inputs to the main computational unit of the cerebellar cortex, the Purkinje cells (PCs) [32,33,34,35,36]. The electrical inputs conveyed by the mossy fiber pathway modulate the most frequent spike type of PCs, the simple spikes (SSs), [37]. The climbing fiber system triggers the second type of PC spike, the relatively infrequent complex spike. Climbing fibers are the axons of inferior olivary neurons, which exhibit characteristic subthreshold oscillations (STO) of their membrane potentials that can result in rhythmic complex spike activity [38,39,40,41,42,43,44]. The finding that those oscillations mostly overlap the theta-band suggests that PCs are particularly suitable to receive inputs at this frequency band [45]. External sensory input, within the theta frequency band, is also particularly effective in modulating the activity of the main PCs target, the cerebellar nuclei neurons (CNNS) [46]. Therefore, ample evidence suggests that the olivo-cerebellar system preferably operates at the theta frequency band. However, whether PC activity is particularly effective in inducing rhythmic behaviors when operating at any specific frequency has not yet been explored.

We hypothesized that stimulation of PCs at the preferred oscillatory frequencies of the climbing and mossy fiber systems would be particularly effective in driving motor output. To test this hypothesis, we used head-fixed, awake mice expressing channelrhodopsin-2 specifically in PCs or in their climbing fiber to perform optogenetic stimulation and extracellular single-unit recordings. Spontaneous or induced whisker movements were monitored with a high-speed camera. During spontaneous whisking, both SS instantaneous frequency and whisker movement exhibited oscillatory behavior with peaks within the theta-band in their power spectrum. Subsequently, we elicited either simple or complex spikes at the frequencies of 0.5, 1, 2, 4, 8, 14, and 28 Hz. We found that 8 Hz is the preferred stimulation frequency at which the biggest movement is induced. Interestingly, oscillatory whisker movements at 8 Hz were generated also when SS bursting was induced at 2 and 4 Hz. Such oscillatory whisker movements were induced exclusively when SS activity was modulated and never via climbing fiber activation.

## 2. Materials and Methods

Further information and requests for resources and reagents should be directed to and will be fulfilled with the Lead Contact, Vincenzo Romano (v.romano@erasmusmc.nl).

### 2.1. Data and code availability

All data are available from the Lead Contact upon request. The custom code complementing BWTT whisker tracking can be obtained via https://github.com/elifesciences-publications/BWTT_PP (accessed on 15 October 2018). Any additional information required to reanalyze the data reported in this paper is available from the lead contact upon request.

### 2.2. Experimental Model and Subject Details

#### Mice

Wild-type C57BL/6J (No. 000664) and transgenic Ai27D (No. 012567) mice were obtained from the Jackson Laboratory. Mice 6–34-weeks-old expressing channelrhodopsin-2 exclusively in their PCs, as described previously in [6], were used in this study and the mice were housed individually in a 12-h light–dark cycle with food and water ad libitum. The ambient housing temperature was maintained at ~25.5 °C with 40–60% humidity. We used 16 mice for the optogenetic stimulation experiments and 53 mice for the 233 PCs and 78 CNNs recordings.

### 2.3. Method Details

#### 2.3.1. Surgeries

For all behavioral and electrophysiological recordings, a custom-made magnetic pedestal was attached to the skull of the mouse above bregma using Super-Bond C&B (Sun medical, Moriyama, Japan) and a craniotomy was made over simplex, Crus1/Crus2 and paramedian lobule. Surgical procedures were performed under isoflurane anesthesia (ISOFLUTEK^®^ 1000, Karizoo, Barcelona, Spain; 2–4% *v/v* in O_2_). Mice were given 5 mg/kg carprofen (“Rimadyl”, Pfizer, New York, NY, USA), 1 µg lidocaine (AstraZeneca, Zoetermeer, The Netherlands), 50 µg/kg buprenorphine (“Temgesic”, Reckitt Benckiser Pharmaceuticals, Slough, UK), and 1 µg bupivacaine (Actavis, Parsippany-Troy Hills, NJ, USA). After 48 h of recovery, mice were habituated to the recording setup during at least two daily sessions of approximately 45 min. In the recording setup, mice were head-fixed with the pedestal and restrained.

#### 2.3.2. Whisker Movement Recording and Tracking

Videography was performed using a high-speed camera (acA640–750 um, Basler Electric, Highland, IL, USA) placed at ~50 cm above the mouse. The whisker movement was captured at a frequency of 1 kHz (1 frame/ms). The whisker movements were tracked as described previously [8] using the BIOTACT Whisker Tracking Tool [47] in combination with custom written code (https://github.com/elifesciences-publications/BWTT_PP, accessed on 12 May 2022). The whisker movements were described as the average angle of all trackable whiskers per frame. For a subset of data, we used a new custom written tracking algorithm: WhiskEras [48], (https://whiskeras.nl/, accessed on 12 May 2022).

#### 2.3.3. Electrophysiology

Electrophysiological recordings were performed in awake mice using quartz-coated platinum/tungsten electrodes (2–5 MΩ, outer diameter = 80 µm, Thomas Recording, Giessen, Germany) placed in an 8 × 4 matrix (Thomas Recording), with an inter-electrode distance of 305 µm. Prior to recording, mice were lightly anesthetized with isoflurane to remove the dura, fix them in the apparatus, and adjust all manipulators. Recordings in right simplex, Crus1/Crus2 and paramedian lobule at a minimal depth of 500 µm began at least 40 min after termination of anesthesia. The electrophysiological signal was digitized at 25 kHz, using a 1–6000 Hz band-pass filter, 22× pre-amplified, and stored using a RZ2 multi-channel workstation (Tucker-Davis Technologies, Alachua, FL, USA). Spikes were detected offline using SpikeTrain (Neurasmus, Rotterdam, The Netherlands). A recording was considered to originate from a single PC when it contained both CSs (identified by the presence of stereotypic spikelets) and SSs, when the minimal inter-spike interval of SSs was 3 ms and when each CS was followed by a pause in SS firing of at least 8 ms. We accepted only those recordings during which the amplitude and the width of the spikes were constant over time. The recordings in which the amplitude or the width of more than three consecutive spikes exceeded three standard deviations (SDs) above or below their average were considered unstable and excluded. In this way, any change in spike rate due to the instability of the recordings was avoided. When these criteria were satisfied, we considered them stable single-unit recordings, and those with a minimum recording duration of 40 s that comprehended whisker movements were selected for further analysis. In total 59 PCs cells fulfilled these criteria.

#### 2.3.4. Viral Injections

AAV9-Syn-ChrimsonR-tdTomato (AAV-ChrimsonR) was obtained from UNC Vector Core, Chapel Hill, NC, USA. All viral vectors were aliquoted and stored at −80 °C until used. To express AAV-ChrimsonR in climbing fibers, 50 µL of AAV9-Syn-ChrimsonR-tdTomato viral vectors was injected in the left inferior olive using the following coordinates: −2.9 mm A–P to lambda, 0.5 mm M–L, and 5.3 mm D–V. After 6 weeks of incubation, a craniotomy (as described above) was performed on the contralateral cerebellar hemisphere (right side). The details of this approach have been described previously [49].

#### 2.3.5. Histology and Microscopy

To verify the site and the size of the viral injection, animals were deeply anesthetized with isoflurane and intraperitoneal injection of pentobarbital sodium solution (50 mg/kg). They were perfused transcardially with saline, followed by 4% paraformaldehyde (PFA) in 0.1 M phosphate buffer (PB, pH 7.4). Brains were removed immediately and post-fixed for an hour in 4% PFA in 0.1 M PB at 4 °C. Fixed brains were placed in 10% sucrose overnight at 4 °C and then embedded in 12% gelatin−10% sucrose. After fixation in 10% formalin for an hour, the blocks were placed in 30% sucrose overnight at 4 °C, 40-µm serial coronal sections were cut with a freezing microtome (SM2000R, Leica, Wetzlar, Germany) and collected in 0.1 M PB. For all immunofluorescence sections, DAPI was used for general background staining. For fluorescence imaging, we took overviews of the brains with a 10× objective on a fluorescence scanner (Axio Imager.M2, ZEISS, Oberkochen, Germany). Images of coronal sections with labelling in the inferior olive are shown in Appendix A.

#### 2.3.6. Optogenetic Stimulation

LED photostimulation (wavelength = 595 nm, M595F2, Thorlabs, Newton, NJ, USA) or (wavelength = 470 nm, M470, Thorlabs, Newton, NJ, USA) was given by a high-power light driver (DC2100, Thorlabs, Newton, NJ, USA) through an optic fiber (400 µm in diameter, Thorlabs, Newton, NJ, USA). The optic fiber was placed on the surface of the paramedian lobule (0.3 mm lateral from the vermal longitudinal sulcus).

### 2.4. Quantification and Statistical Analysis

#### 2.4.1. Experimental Paradigm

The duration of each single LED pulse was always 0.02 s (20 ms). Several Inter-Pulse Intervals (IPI, time between onset of two consecutive stimuli) were used to test the effect of different stimulation frequencies. We have tested seven trains of pulses at the following stimulation frequencies: 0.5 Hz (IPI = 2 s), 1 Hz (IPI = 1 s), 2 Hz (IPI = 0.5 s), 4 Hz (IPI = 0.25 s), 8 Hz (IPI = 0.125 s), 14 Hz (IPI = 0.077 s), and 28 Hz (IPI = 0.035 s). Hence, within each train of stimuli, the interval in which the LED was off depended on the stimulation frequency and was equal to the IPI minus 0.02 s. To maintain the overall duration of the trains of stimuli at each frequency (~20 s), we varied the number of stimuli for each train. The protocol consisted of 10 stimuli at 0.5 Hz, 40 stimuli at 2 Hz, 280 stimuli at 14 Hz, 140 stimuli at 8 Hz, 660 stimuli at 28 Hz, 80 stimuli a 4 Hz, and 20 stimuli at 1 Hz. The interval between subsequent trains of stimuli (one per each stimulation frequency) was ~10 s. Therefore, we alternated ~20 s of stimulation to ~10 s of resting period. To check whether stimulating a second epoch with a similar frequency would result in a different type of movement, 0.5 and 14 Hz stimulations were given twice.

Data analysis was performed in Matlab R2021a using custom-written scripts unless noted otherwise.

#### 2.4.2. Simple Spike Analysis

Raw PC traces were spike-sorted using SpikeTrain (Neurasmus BV, Rotterdam, The Netherlands), resulting in spike trains (Figure 1), which were converted into logical SS vectors. SS instantaneous frequencies were calculated by convolving a logical SS vector with a multiple of Gaussian kernel probability density function. The kernel had a width of 100 data points, corresponding to 0.1 s, and was created with the following gaussian:e^((−x2)/(2n2))^(1)

The multiple of Gaussian kernel was created by averaging two kernels with values n = 8 and n = 20.

As we were interested in whisking frequencies and to cancel out the effect of pink (or 1/f) noise, the SS instantaneous frequency was bandpass-filtered between 3 and 25 Hz using a Butterworth bandpass filter.

#### 2.4.3. Whisker Analysis

Raw whisker angle vectors were low-pass filtered with a cutoff frequency of 30 Hz using a Butterworth low-pass filter. Whisker traces were analyzed to detect protractions and retractions, which were used to define periods of movement. Whisker movement with a minimum change in angle of ±4° was considered a protraction or retraction, respectively. For every epoch, whisker traces were aligned to the onset of stimulation to create the mean whisker trace.

Mean whisker traces were created by calculating the mean of all mean whisker traces. Shaded areas indicate SEM when data consists of average from averages. Otherwise, shaded areas depict SD.

The amplitude of whisker movement was calculated as the maximum value of the mean whisker trace minus the minimum of the mean whisker trace from the onset of the stimulus to the onset of the subsequent stimulus. For frequencies below 4 Hz, the amplitude was calculated in a time window of 250 ms after the onset of the stimulus to exclude spontaneous whisks in between stimuli. 

The whisker amplitudes used to investigate entrainment were calculated as the maximum value of the whisker trace minus the minimum value of the whisker trace between two consecutive stimuli for the first 15 stimuli. Slopes were calculated by plotting a straight line through the amplitudes calculated per mouse, resulting in a slope for every stimulation frequency for every mouse. Average slopes per frequency were calculated for both PC and climbing fiber stimulation.

Scalograms were created using continuous wavelet transform (CWT) from the Matlab wavelet toolbox. A regular fast Fourier transform creates a power spectrum in the frequency domain. These data, however, lose all information from the time domain. To keep information in both frequency and time domain, continuous wavelet transform (CWT) was used with the Morlet wavelet.

## 3. Results

### 3.1. Oscillatory Simple Spike Frequency and Whisking Kinematics

To investigate whether, under our experimental condition, the firing pattern of PC oscillates in a specific frequency band [45], we recorded 59 PCs from awake, freely whisking mice. We converted the SS train into an instantaneous frequency vector and determined oscillation frequencies using power spectrum analysis (Figure 1A–D). The majority of the power of the instantaneous SS oscillation ranged between 3 and 20 Hz (Figure 1D), while the peak of maximal power was always between 5–10 Hz (mean peak = 5.6 Hz, Figure 1E). Next, we analyzed the oscillation of the voluntary whisker movement. In line with previous works [50,51,52], the majority of whisker movements oscillate at ranges between 3–25 Hz (mean peak = 7.2 Hz, Figure 1F–H). A cycle of whisker oscillation was composed of a protraction and a retraction with amplitude, duration, and max velocity that ranged from 8–14°, 35–51 ms, and 0.40–0.68°/ms, respectively (Figure 1I). These two results indicate that the SS activity oscillates at frequencies that partially overlap with the frequency of whisking oscillation.

### 3.2. Optogenetic Stimulation of Purkinje Cells can Induce Specific Patterns of Whisker Movement

Before testing different frequencies of PC stimulation, we compared the whisker movement induced by SS modulation versus climbing fiber stimulation (Figure 2). To modulate SS activity, we used a genetic mouse model expressing channelrhodopsin specifically in PCs [6,53]. Additionally, AAV-ChrimsonR virus was injected into the inferior olive to allow for the modulation of climbing fiber activity by stimulating the contralateral cerebellar cortex [49], (Appendix A). We applied optogenetic stimulation to the paramedian lobule, which has been associated with strong oscillatory activity during voluntary movements [54]. A train of 4 Hz light stimulation (80 pulses, lasting 20 ms each) to either PCs or climbing fibers resulted in different patterns of whisker response. While stimulation of climbing fibers resulted in one individual cycle of whisker movement (i.e., protraction and retraction) per light pulse, two consecutive whisker cycles followed each SS stimulation (Figure 2D–E). Therefore, whisker oscillation at a frequency (~8 Hz) that is a multiple of the stimulation frequency (4 Hz) can be induced by SS, but not by climbing fiber modulation.

### 3.3. Purkinje Cell Activity Induces Whisker Movements at the Preferred Frequency of 8 Hz

To test which stimulation frequency generates maximal movement, we applied 0.5, 1, 2, 4, 8, 14, and 28 Hz light pulses to cover the predominant range of voluntary whisking frequencies (Figure 3). Similar to 4 Hz stimulation (Figure 2D), SS stimulation at 0.5, 1, and 2 Hz resulted in whisker fluctuation at higher frequencies (Figure 3A–E). Power spectrum analysis (Figure 3C, see also Appendix A) revealed that during 2, 4, and 8 Hz stimulation, the most abundant whisking frequency was ~8 Hz for SS stimulation (7.1, 7.9, and 7.9 Hz whisker movement for stimulation at 2, 4, and 8 Hz, respectively), but not for climbing fiber stimulation. This is because climbing fiber stimulation always induces only one cycle of whisker movement per stimulus. Next, whisker amplitudes were extracted to see which frequency results in the biggest whisking amplitude. The highest amplitude of whisker movement around the stimulus is reached during 8 Hz stimulation for both climbing fiber and SS stimulation (Figure 3E). Based on these results, we concluded that 8 Hz is the preferred PC frequency to generate whisker movement.

### 3.4. Simple Spike, but Not Climbing Fiber Stimulation, Entrains Whisker Movement

To test whether recurrent loops undergo entrainment processes that make whisker movement amplitude grow over time, we calculated the amplitude of movement between two consecutive stimuli for the first 15 stimuli of each train of stimulation (one per each frequency), (Figure 4). To exclude spontaneous whisks between larger intervals, for stimulation frequencies of 0.5, 1, and 2 Hz, we focused on the maximum amplitude in the first 250 ms after the stimulus. During SS stimulation, an increase in amplitude over time could be observed for all frequencies (Figure 4A). To quantify the increase of amplitude, we calculate the slope of the linear fit between amplitude per stimulus (see also methods). Such an increase in amplitude was not visible during climbing fiber stimulation (Figure 4B). Comparing all frequencies, the slope was higher during SS stimulations compared to climbing fiber stimulations (paired sample *t*-test, *p* = 0.005). If the system undergoes oscillatory entrainment, there is the possibility that whisking oscillation persists after the end of the stimulation at the specific frequency of the last few stimuli. To test this possibility, we compared the power spectra of the whisker movements 1 s before and 1 s after the last stimulus of each stimulus train (Figure 5). While whisker movement often continued after the end of the stimulation, especially for SS modulation, the frequency of these movements never reflected the previous stimulation frequency. We concluded that SS stimulation induces entrainment leading to a higher amplitude whisker response over time and the modulation of PCs is required to sustain this entrainment.

## 4. Discussion

We tested whether patterns of PC activity, at any specific frequency, are particularly effective for rhythmic whisking induction. We found that 8 Hz stimulation induces the highest amplitude whisker movement during both SS and climbing fiber stimulation. Interestingly, 8 Hz whisker movement was induced also at 2 and 4 Hz stimulation of SSs, but not of climbing fibers. Therefore, 8 Hz appears to be the preferred frequency at which the olivocerebellar system generates whisker movement. 

### 4.1. Simple Spike Activation of Reverberating Circuits Sustaining Rhythmic Whisks

Our findings show that the oscillatory properties of the olivocerebellar system, that have been extensively studied in vitro and in an anesthetized preparation [21,28,31,33,34,38,39,41,45,46,55,56], can be read out on rhythmic whisker movement by optogenetically manipulating the PCs. Our approach provides evidence for the presence of reverberating loops that generate oscillatory cycles that depend on the SS activity. When SS bursts were induced at 2 and 4 Hz, two cycles of whisker movement were generated per stimulus. Conversely, only one cycle of whisker movement was induced upon climbing fiber activation. Hence, climbing fiber stimulation was effective in inducing whisker movements but did not generate reverberating signals. This difference can be explained by the different effects of the two types of stimulation on the downstream CNNs. The pulse of 20 ms stimulation induces a train of multiple SSs (~5 spikes) in L7^Cre^;Ai27 mice, resulting in an inhibition–disinhibition rebound firing pattern in the CNN [6]. Climbing fiber stimulation, in contrast, induced a pattern consisting of disinhibition of CNNs in absence of the preceding inhibition [37]. Therefore, the alternation of inhibition–disinhibition in CNNs may be required to generate the oscillatory whisking cycles at 8 Hz, with stimuli at 2 and 4 Hz. The resulting power spectra peaked at 8 Hz, indicating that for each pulse of SS stimulation, the whisker muscles received motor commands ~125 ms apart. Therefore, the cerebellar output activated one or multiple brain circuits in which the neuronal signal reverberated in a loop with a period of ~125 ms. This possibility can explain also why the biggest whisker movements are induced when stimuli were given every 125 ms (i.e., 8 Hz). The temporal summation of the reverberating inputs could have caused neuronal resonance that resulted in the largest amplitude of whisker oscillation. Future experiments can address the question of where such temporal summation takes place.

### 4.2. Multiple Cerebellar–Whisker Pathways 

Whisker motor neurons, located in the facial nucleus [57,58], receive projections from multiple premotor whisker nuclei [22,59]. It has been proposed that rhythmic whisker movements are generated by a central pattern generator, located in the intermediate reticular formation [52,60]. Besides the reticular formation, cerebellar output targets multiple premotor whisker regions such as the red nucleus, superior colliculus, and spinal trigeminal [25,59]. Therefore, PC stimulation could affect multiple parallel pathways responsible for oscillatory whisker movement. Other pathways could also involve cerebrocerebellar loops [14,61]. Electrophysiological recordings and optogenetic manipulation of the pre-motor or neocortical whisker regions could reveal the specific circuit in which specific patterns of the cerebellar output can reverberate.

### 4.3. The Olivocerebellar Contribution to Rhythmic Whisking 

The olivocerebellar system is thought to be a modulator of the ability of the motor cortex to generate rhythmic movements such as whisking [24]. Our results suggest that the olivocerebellar system could constrain rhythmic whisking at the specific frequency of 8 Hz. Here, however, we tested only two specific patterns of stimulation (20 ms pulses of climbing fiber and SS stimulation). Therefore, we cannot exclude that different patterns of PC activity can also constrain rhythmic whisking at other frequencies. Remarkably, the olivocerebellar system is characterized by a feedback loop by which PCs control their own climbing fiber discharges [62]. Within this loop, the cerebellar output returns to the cerebellar cortex via climbing fiber activity generated by the di-synaptic disinhibition of olivary neurons. The timing of such recurrent activity can slightly change depending on the duration of the SS bursts [62]. This change in the period of the recurrent activity, in principle, could affect the frequency at which the motor output is constrained by the olivocerebellar activity. Alternatively, different frequencies of whisker oscillations may be regulated by different extra-olivocerebellar loops. Besides the direct cerebellar-olivary projections there are other di-synaptic connections between cerebellar nuclei and inferior olive that could mediate the return of the cerebellar output to the cerebellar cortex via climbing fiber activity. Brain structures such as the meso-diencephalic junction and superior colliculus are known to receive inputs from cerebellar nuclei and excite inferior olivary nuclei [23,63]. The presence of these parallel recurrent pathways makes the olivocerebellar system a particularly suitable neuronal substrate for sustaining recurrent activity. In addition, the high tonic firing of PC and CNN can undergo bidirectional modulation and fluctuations around their mean value [37].

We propose that recurrent activity traveling across the above-mentioned parallel loops can induce periodic fluctuations of PC and CNN firing that generate transient cycles of activity oscillations in the downstream pre-motor targets generating the rhythmic movement.

### 4.4. Olivocerebellar Rhythmic Activity beyond Mouse Whisking 

Neuronal oscillations are present across several brain areas and the fact that they are phylogenetically preserved suggests that they are functionally relevant [64]. Gaining knowledge on the specific frequency at which specific neuronal types exhibit their propensity to oscillate could be beneficial for further improving treatment approaches such transcranial or deep brain stimulation [65]. In this study, we demonstrated that PC activity is particularly effective in producing motor output when stimulated at 8 Hz. Whether stimulation at this frequency could be particularly effective for the treatment of diseases remains to be elucidated. It has been reported that deep brain stimulation of the meso-diencephalic junction, a brain area that is directly connected to the olivo-cerebellar system, abolished the symptoms of posttraumatic movement disorder [66]. Conversely, the impact of transcranial direct current stimulation (tDCS) and transcranial alternating current stimulation (tACS) on cerebellar disorders, remain unclear [65]. This difference could be because transcranial brain stimulation is weaker than deep brain stimulation. Therefore, maximizing the effectiveness of tACS, by using the most effective stimulation frequency, could improve its impact on recovery from traumatic brain injuries and strokes [67,68].

## Figures and Tables

**Figure 1 biomedicines-10-01831-f001:**
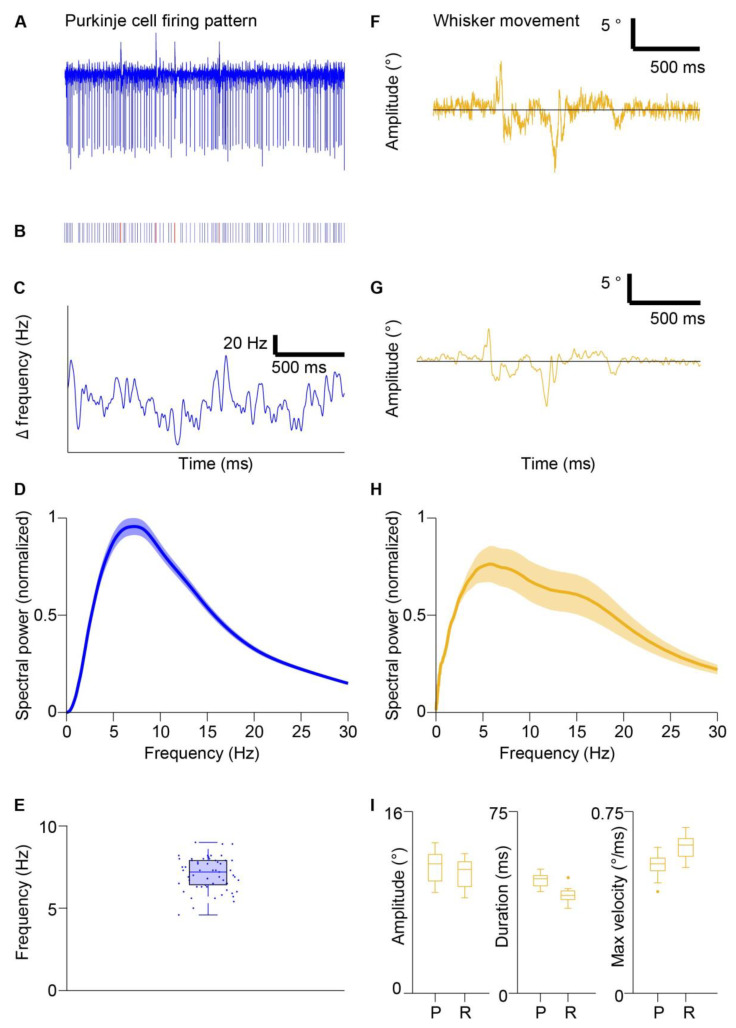
Oscillatory frequencies of simple spiking and whisking kinematics. (**A**) shows an example of raw trace of a PC recording before spike sorting. (**B**) shows the spike train of SSs (in blue) and complex spikes (in red) extracted from the raw trace of (**A**). (**C**) shows corresponding instantaneous frequency from (**A**,**B**). (**D**) shows a power spectrum of SS instantaneous frequency during periods of movement, calculated with the continuous wavelet transform. The line displays the mean and shaded areas display ± SD, n = 59 cells. (**E**) shows a boxplot of frequencies with a maximum power of power spectra summarized in (**D**), 7 ± 1 Hz (mean ± SD), n = 59 cells. The value of each cell is shown as a dot. (**F**) shows an example of raw whisker trace. (**G**) shows a filtered whisker trace from (**F**). (**H**) shows a power spectrum of whisking during periods of movement, n = 13 mice. Line displays mean and shaded areas display ± SD. (**I**) shows a boxplot with amplitude, duration, and maximum velocity of protractions (P) and retractions (R) during voluntary whisking. Boxplots indicate mean and interquartile ranges. The dots indicate the outliers.

**Figure 2 biomedicines-10-01831-f002:**
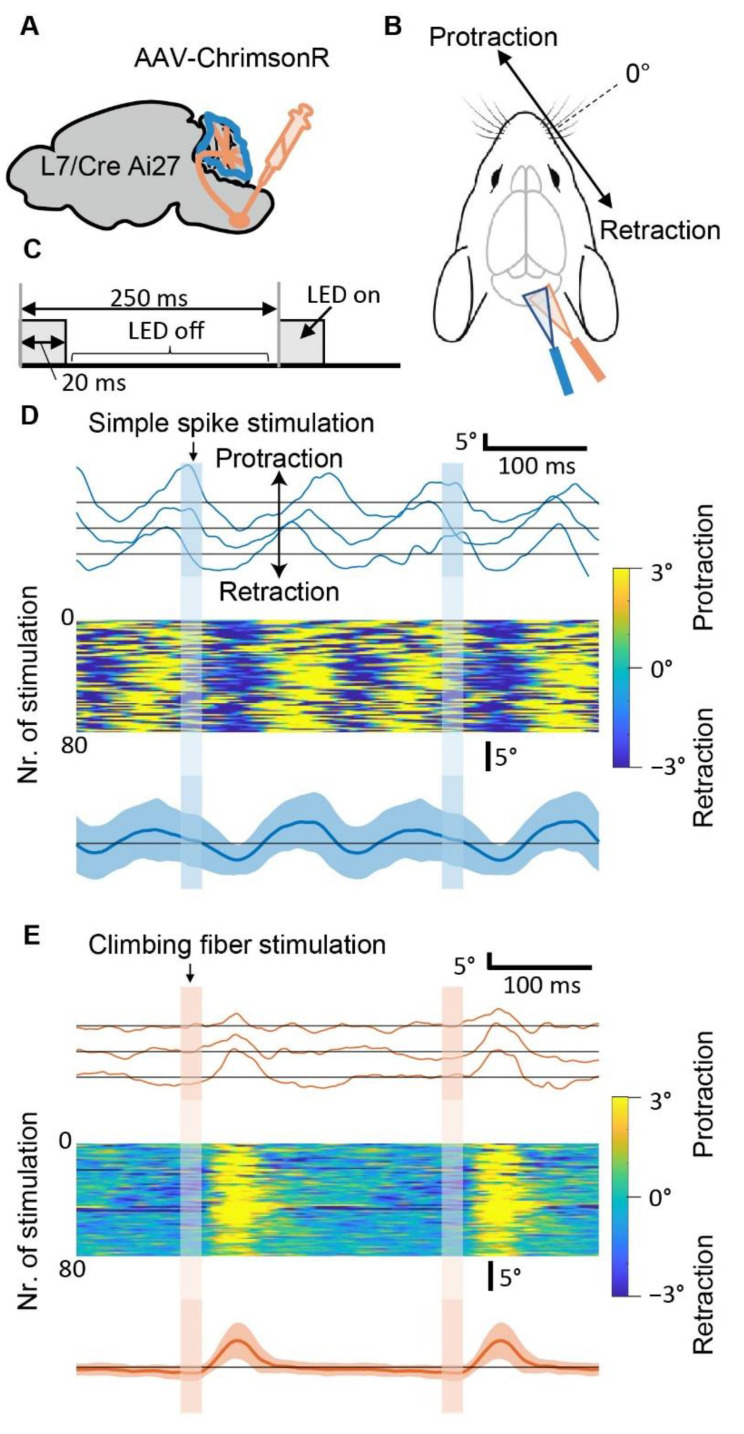
Optogenetic stimulation of Purkinje cells can induce specific patterns of whisker movement. (**A**) Schematic of AAV-ChrimsonR injection in the inferior olive in L7^Cre^;Ai27 mice. The blue area highlights the cerebellar cortex, in which SS rates increase upon stimulation by a blue LED light. The climbing fibers from the inferior olive are sensitive to orange LED light. (**B**) Schematic of the experiment. Cerebellar lobules are stimulated with either an orange or blue LED, during which whisker movements are captured with a high-speed camera (1 kHz). (**C**) Schematic of the stimulation paradigm. (**D**) The upper panel shows whisker traces of three consecutive stimulations of the paramedian lobule at 4 Hz (blue lines). Below, are shown all 80 whisker traces of this experiment and the resulting mean whisker movement for an exemplary mouse. Shaded areas indicate mean ± SD. (**E**) is similar to (**D**) but with orange LED stimulation of the climbing fibers (orange lines).

**Figure 3 biomedicines-10-01831-f003:**
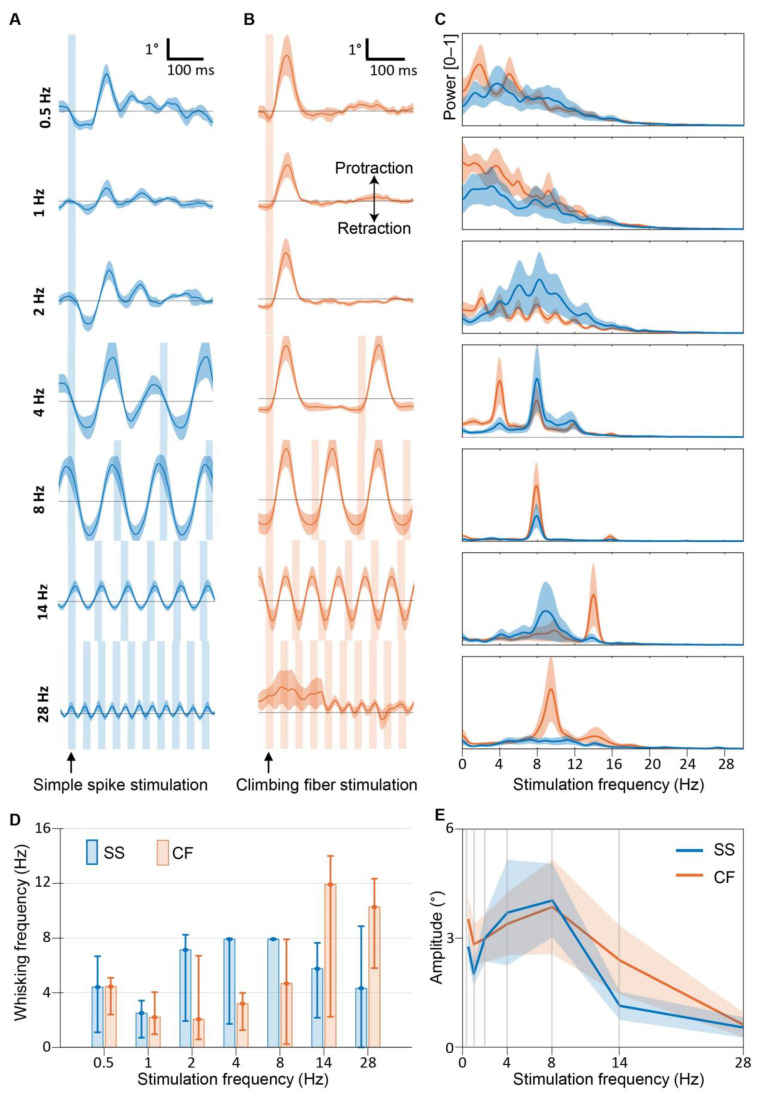
Purkinje cell stimulation induces whisker movements at the preferred frequency of 8 Hz. (**A**) shows mean ipsilateral whisker traces during SS stimulation (blue light) in the paramedian lobule on the right hemisphere. Vertical blue bars represent stimuli, lines represent mean, and the shaded areas display ± SEM (n = 6 mice). (**B**) is similar to (**A**), but for climbing fiber stimulation (n = 8 mice). (**C**) shows normalized whisker power spectra during the stimulation. The stimulation frequencies of each plot is the same indicated on the left part of panel (**A**). Also here, the lines represent the mean, and shaded areas indicate ± SEM. (**D**) shows a bar plot of the frequency at which the power spectra peaked when different frequencies were given (indicated on the x axes). (**E**) shows a psychometric curve of the amplitude of the whisker movement relative to the stimulation frequency. Note that the whisker movements were maximal within the theta-band (4–8 Hz).

**Figure 4 biomedicines-10-01831-f004:**
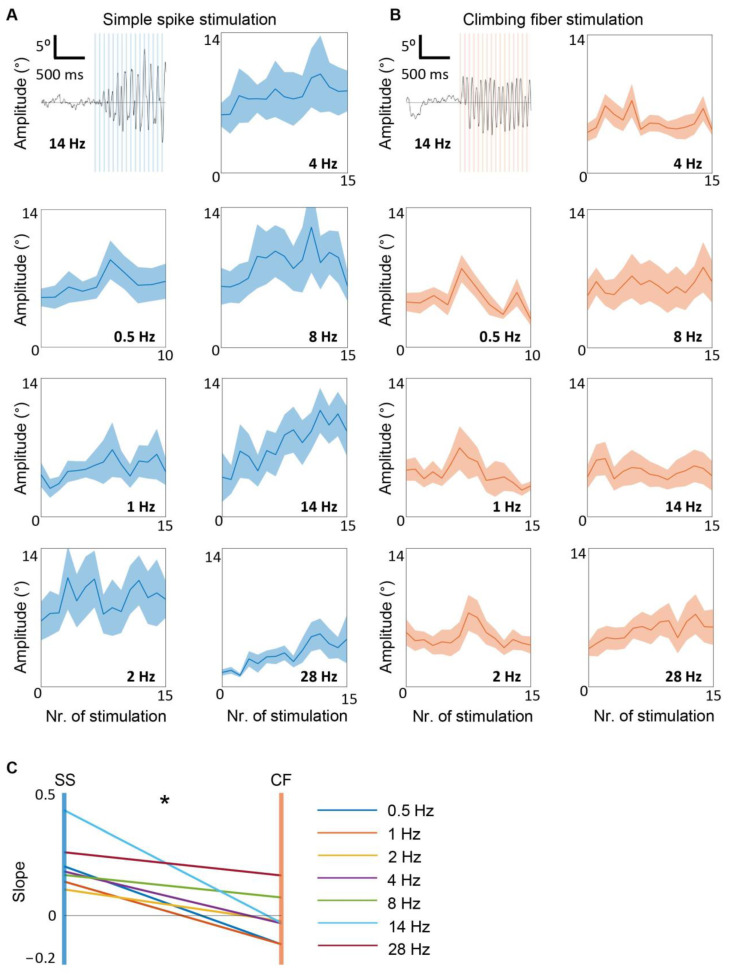
Simple spike, but not climbing fiber stimulation entrains whisker movement. (**A**) shows an example of whisker raw trace during the first 15 pulses of SS stimulation at 14 Hz (upper left). Vertical blue bars represent stimuli. Note how the amplitude gradually increases over time. The mean whisker amplitude during the first 15 stimulations is shown for 1, 2, 4, 8, 14, 28. For 0.5 Hz are shown only for 10 stimuli (corresponding to the total number of stimuli given at that frequency). (**B**) is similar to (**A**), but for climbing fiber stimulation. (**C**) shows that the values of the slopes during SS stimulation were higher than during climbing fiber stimulation (*p* = 0.0047, paired sample *t*-test). The symbol * indicates that *p*-value is <0.05.

**Figure 5 biomedicines-10-01831-f005:**
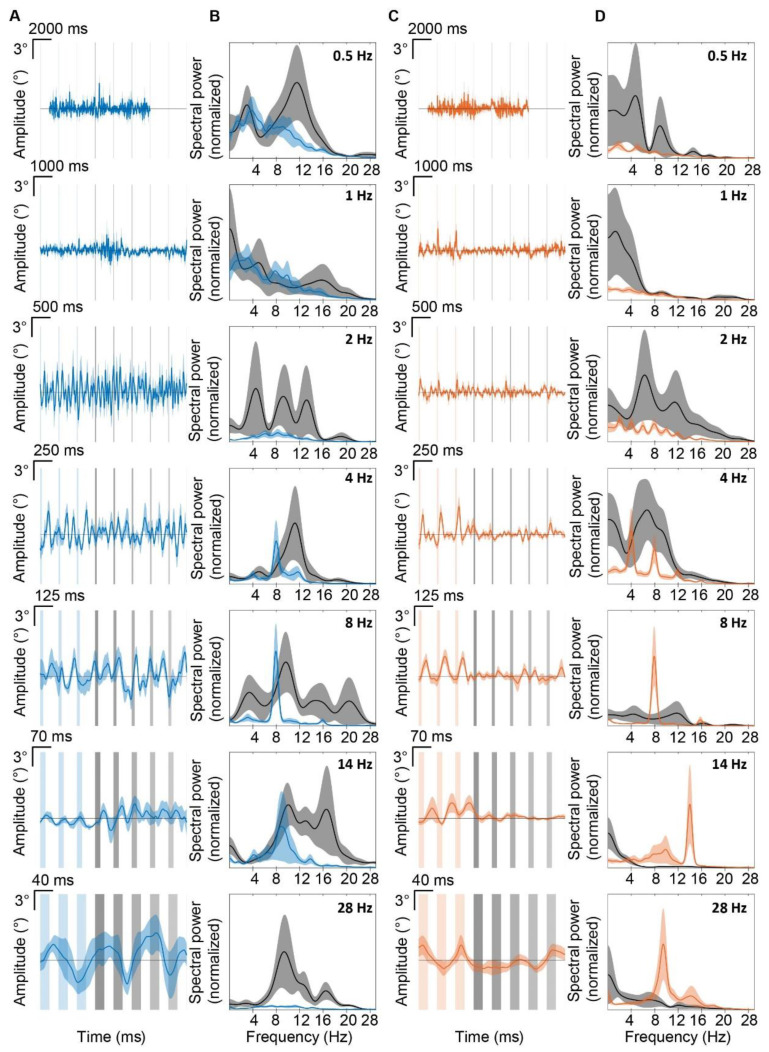
Entrainment of whisker movement is sustained by Purkinje cell activity. (**A**) shows mean whisker traces before and after the end of a train of stimuli of PC stimulation (left). Stimuli are indicated by blue bars. Grey bars indicate the timing at which subsequent movement could be expected if the whisker would be keeping oscillating at the frequency of stimulation (left). (**B**) shows power spectra during the period of SS stimulation (blue lines) and the power spectra during 1000 ms after the last stimulus (grey lines). Shaded areas indicate mean ± SEM. (**C**,**D**) are the same as (**A**,**B**), but for climbing fiber stimulation (orange lines). Note that the power spectra are different for the period before and after the end of the stimulation.

## Data Availability

Further information and requests for resources and reagents should be directed to and will be fulfilled with the Lead Contact, Vincenzo Romano (v.romano@erasmusmc.nl).

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
