# Peer review of "Purkinje Cell Activity Resonation Generates Rhythmic Behaviors at the Preferred Frequency of 8 Hz"

_biomedicines, 2022, doi:10.3390/biomedicines10081831_

Round 1

Reviewer 1 Report

Interesting paper looking at the theta frequency in Purkinje cells and role in rhythmic behavior. 

Spike bursting was induced at 2 and 4hZ but not via climbing fibers. 

The main limitation is relationship to clinical applicability. This paper should be expanded to discuss implications for recovery from TBI (PMID: 24448183), stroke, and treatment approaches for neurodegenerative disease. 

If the above is addressed, reference included, and concepts expanded then the paper could be of interest. 

Author Response

Dear reviewer, 
Thank you for your positive response to our paper “Purkinje cell activity resonation generates rhythmic behaviors at the preferred frequency of 8 Hz”, by Staf Bauer, Nathalie van Wingerden, Thomas Jacobs, Annabel van der Horst, Peipei Zhai, Jan-Harm L. F. Betting, Christos Strydis, Joshua J White, C. I. De Zeeuw , and Vincenzo Romano.
We felt that the comments were constructive and helpful for improving the manuscript, and we have therefore acted upon them accordingly. Specifically, we have expanded the discussion to include implications for treatment approaches as suggested by adding a dedicated section in the discussion that includes the relevant references. 

Reviewer 2 Report

The manuscript submitted by Staf Bauer and coauthors demonstrates the correlation between the frequency of optogenetic stimulation of Purkinje cells and behavioral reactions. The experiments were performed at high methodological level. The work can be recommended for the publication after minor text revisions. - The histology is described in materials and methods section as a separate paragraph, but the results of these experiments are not shown in the main text or supplementary. - Please, clearly describe the duration of stimulations (pulses or LED-off periods) for each frequency.

Author Response

Dear reviewer, 
Thank you for your positive response to our paper “Purkinje cell activity resonation generates rhythmic behaviors at the preferred frequency of 8 Hz”, by Staf Bauer, Nathalie van Wingerden, Thomas Jacobs, Annabel van der Horst, Peipei Zhai, Jan-Harm L. F. Betting, Christos Strydis, Joshua J White, C. I. De Zeeuw , and Vincenzo Romano.
We felt that the comments were constructive and helpful for improving the manuscript, and we have therefore acted upon them accordingly. Specifically, we added a new supplementary figure (Figure S1) that shows the results of the histology of all mice that received the viral injection. Finally, we expanded the description of the duration of each stimulation specifying the intervals of LED-off periods. 

Round 2

Reviewer 1 Report

Accept